# Prevalence and psychosocial impact of atopic dermatitis in Bangladeshi children and families

**Courtney J. Pedersen[1], Mohammad J. Uddin[2], Samir K. Saha[2], Gary L. Darmstadt®[3]\***

**1** Stanford University School of Medicine, Stanford, California, United States of America, **2** Child Health Research Foundation, Dhaka, Bangladesh, **3** Department of Pediatrics, Stanford University School of Medicine, Stanford, California, United States of America

\* gdarmsta@stanford.edu

## Abstract

### Background

Skin conditions are the fourth leading cause of nonfatal disease globally, with atopic dermatitis (AD) a major and rising contributor. Though atopic dermatitis (AD) is rising in prevalence, little is known about its psychosocial effects on children and families in low- and middle-income countries (LMICs).

### Methods

We conducted a community-based, cross-sectional survey of 2242 under-5 children in rural Bangladesh using the International Study of Asthma and Allergies in Childhood (ISAAC) questionnaire to measure AD prevalence and severity, the Patient-Oriented Eczema Measure (POEM) to measure severity, and the Infants' Dermatitis Quality of Life index (IDQoL) and Dermatitis Family Index (DFI) to measure quality of life.

### Findings

Overall AD prevalence in under-five children was 11.9% [95% confidence interval (CI) 10.6–13.3%]. Prevalence was highest in children age 30–35 months [16.2% (95% CI 11.4–21.0)]. IDQoL was significantly higher in males (2.67) vs. females (1.95, p = 0.015), the lowest (3.06) vs. highest (1.63) wealth quintile (p<0.001), and among mothers with < primary (2.41) vs. > secondary (1.43) education (p = 0.039). POEM severity was correlated with IDQoL (r = 0.77, p<0.001) and DFI (r = 0.56, p<0.001). Severe disease as rated by caretakers was correlated with POEM (r = 0.73, p<0.001), IDQoL (r = 0.82, p<0.001) and DFI (r = 0.57, p<0.001).

### Conclusions

Severe AD significantly affects quality of life for children and families in Bangladesh. As access to healthcare expands in LMICs, identification and treatment of both the medical and psychosocial morbidities associated with the disease are needed.

**Data Availability Statement:** The data used for this paper contains Protected Health Information. Stanford University obtained this data from Dr. Samir Saha (a co-author on this paper), who manages the Demographic Surveillance Site where

the authors conducted data collection. The Institutional Review Board in the Research Compliance Office, Stanford University, for protocol #41405 imposed the data sharing restrictions. The data are available through a data sharing agreement with co-author, Dr. Samir Saha (samir@chrfbd.org), upon reasonable request. Authors may also contact the Child Health Research Foundation about data sharing: Mohammad Shahidul Islam Senior Research Investigator Child Health Research Foundation shahidul@chrfbd.org.

**Funding:** Dr. Pedersen received a Medical Scholars grant from the Stanford University School of Medicine and a Benjamin H. Kean Travel Fellowship from the American Society of Tropical Medicine and Hygiene. The REDCap platform services are made possible by the Stanford University School of Medicine Research Office and are subsidized by the National Center for Research Resources and the National Center for Advancing Translational Sciences, National Institutes of Health. The data content is solely the responsibility of the authors and does not necessarily represent the official views of the NIH. The funders played no role in the study design, data collection and analysis, decision to publish, or preparation of the manuscript.

**Competing interests:** The authors have declared that no competing interests exist.

## Introduction

Skin conditions are the fourth leading cause of nonfatal disease globally [1]. A key contributor to the burden of skin disease is atopic dermatitis (AD)–a chronic, inflammatory skin disorder that begins before age five in 90% of cases and is estimated to affect nearly one in five children [2,3]. AD can have severe psychosocial impacts on children and their families; in one study AD was second only to cerebral palsy in negatively impacting children's quality of life [4]. Children with AD can experience physical distress, mood changes, sleep dysfunction, and behavioral problems [5–8]. Families experience the financial burden of treatment and impacts such as sleep deprivation, anxiety, depression, social isolation, and marital problems [9–12].

Data from an international study using validated questionnaires found that AD prevalence is increasing in low- and middle-income countries (LMICs), especially among children aged 6 to 7 years old [13]. However, as healthcare access improves in LMICs, increased care-seeking and case detection and reporting may also contribute to higher prevalence estimates.

Psychosocial impacts associated with AD are well-known in high-income countries (HICs), including in Asia [14], but there have been no such publications from Bangladesh. The present study aimed to measure the prevalence of AD and examine its psychosocial impacts on children and their families.

## Methods

We conducted community-based, cross-sectional weekly household surveillance visits by community health workers (CHWs) to identify households with under-five children in the 300,000 population Mirzapur Demographic Surveillance Site (MDSS) in a rural sub-district north of Dhaka from November 2017 to April 2018. Details of the study design and conduct have been reported previously [15]. A sample size of 2,149 under-five children was estimated to provide a 95% confidence interval with 1% precision assuming 6.5% prevalence of AD [16]. All 156 villages in the MDSS were subdivided into 110 clusters each and ten clusters were randomly chosen to be included in the study, approximating a population-based sample. Questionnaires were administered by CHWs to the main caretaker of the child, typically the female head of household, who responded on behalf of all children under the age of 5 within her household. CHWs were bilingual and educated to grade 12 or higher. They received two days of classroom instruction on study aims and research protocol, performed one day of field practice with observation and feedback, and received a refresher training one month after the initial training session. Supervisors attended the training sessions and monitored the CHWs weekly. Demographic data were obtained from the MDSS data repository linked to census-based individual and household identification numbers of study participants.

### ISAAC

We used the International Study of Asthma and Allergies in Childhood (ISAAC) Phase 3 core symptom questionnaire to assess the prevalence and severity of AD since it is the most widely used epidemiologic measure of AD and contains a severity measure. ISAAC has been validated in 6–7 and 13–14 year-old children and has been used but not validated in children as young as 2 years [17]. Our methods of ISAAC administration and comparability to the U.K. Criteria [18–20], an assessment tool that has been validated in infants [21,22] and young children [20], were reported previously [15]. AD was defined as an itchy rash at any time coming and going for at least 6 months, that had at any time affected the folds of the elbows, behind the knees, in front of the ankles, under the buttocks, or around the neck, ears or eyes in the past 12 months. Severe AD was defined as being kept awake one or more nights per week on average by this itchy rash in the past 12 months. While previous surveys with ISAAC have not included

infants, if the child was under the age of 6 months we considered any presence of rash as affirmative and determined a positive screen by an affirmative response to the question regarding the anatomical location of the rash.

## Patient-Oriented Eczema Measure (POEM)

POEM is a seven-item, validated questionnaire [23–25] that measures the frequency of child symptoms including itch, sleep disturbance, bleeding, weeping or oozing, and cracked, dry, and flaking skin. The main caretaker, typically the mother of the child, responded on behalf of the child. Each question was scored from 0 = 'No days' to 4 = 'Every day' for a maximum score of 28. We used previously validated score bands to categorize participants: clear (score 0–2), mild (score 3–7), moderate (score 8–16), severe (score 17–24), and very severe (score 25–28) [26].

## Psychosocial measures

**Infants' Dermatitis Quality of Life Index (IDQoL).** The IDQoL is a validated questionnaire [5,27] that consists of 10 items regarding physical symptoms, disturbance of normal activities and sleep, and problems with bathing and dressing. Each item was scored 0–3 with a maximum score of 30, where higher scores represent a poorer quality of life [27], however, there are not validated score bands. An additional question was scored separately regarding the severity of dermatitis from 0 = 'None' to 4 = 'Extremely severe' as perceived by the caretaker.

Both the POEM and IDQoL are recommended as core instruments for AD research by the Harmonising Outcome Measures for Eczema international working group [28].

**Dermatitis Family Impact (DFI).** The DFI is a 10-item questionnaire, validated for use in families of children aged 6 months to 10 years old, that measures the impact of the child's AD on the family unit over the past 7 days [29]. The DFI correlates well with the IDQoL [29,30]. Each item was scored from 0–3 for a maximum score of 30. Higher scores represent a greater impairment of the family as a result of AD. No validated score-banding system exists for the DFI.

All questionnaires were translated from English to Bangla by study physicians, pilot tested in the community, and adjusted after discussions with CHWs per ISAAC Phase 3 Manual recommendations [31]. Question 6 and question 4 in the IDQoL and DFI, respectively, were modified by removing the example of swimming as it was not culturally relevant. Question 5 in the DFI was modified from asking about "time spent on shopping for the family" to "time spent going to the market."

## Data analysis

Study data were collected and managed using REDCap [32]. All statistical tests were performed using SAS software (SAS Institute Inc., Cary, NC, USA). Asset-based wealth quintiles [33] were calculated using the entire MDSS, of which our sample was a subset. AD prevalence was calculated using the total number of individuals in each age group for the denominator for that age group; clustering by region or village was not taken into account. Missing data were eliminated using pairwise deletion and the updated denominators are listed in Table 1. The corresponding 95% confidence intervals were calculated using a binomial test. We used nonparametric tests including the Mann-Whitney U Test and the Kruskall-Wallis test to analyze IDQoL and DFI scores, and the Spearman's rank order correlation coefficient to measure associations between ordinal and continuous variables.

**Table 1. Demographic description of the sample, N = 2242.**

| Variable | % (n) |
|---|---:|
| **Age** in months, mean (standard deviation) | 28.8 (16.9) |
| < 6 months | 10.5% (235) |
| 6–11 months | 9.6% (215) |
| 12–17 months | 10.7% (239) |
| 18–23 months | 10.4% (234) |
| 24–29 months | 10.3% (230) |
| 30–35 months | 10.2% (228) |
| 36–41 months | 10.2% (229) |
| 42–47 months | 9.7% (217) |
| 48–54 months | 10.8% (242) |
| 54–59 months | 7.7% (173) |
| **Sex** | |
| Male | 50.7% (1136) |
| Female | 49.3% (1106) |
| **Household religion, N = 2018** | |
| Muslim | 86.6% (1747) |
| Hindu | 13.4% (271) |
| **Highest education (Maternal), N = 2172** | |
| Less than primary | 50.2% (1091) |
| Less than secondary | 36.5% (793) |
| Secondary or higher | 13.3% (288) |
| **Highest education (Paternal), N = 1745** | |
| Less than primary | 49.0% (855) |
| Less than secondary | 33.0% (576) |
| Secondary or higher | 18.0% (314) |
| **Wealth quintile, N = 1968** | |
| Poor | 5.6% (111) |
| Lower middle | 14.6% (288) |
| Middle | 22.0% (432) |
| Upper middle | 24.9% (490) |
| Wealthy | 32.9% (647) |

## Ethical approval

Ethical approval was obtained from the Institutional Review Boards at the Stanford University School of Medicine (protocol #41405) and the Bangladesh Institute of Child Health in Dhaka. Verbal informed consent was obtained from male or female heads of household.

## Results

We surveyed 2068 households, with no refusals, and enrolled 2242 under-five children [mean age 28.8 months, standard deviation (SD) 16.9 months] including 1136 (50.7%) males and 1106 (49.3%) females (Table 1). Children were categorized into 6-month age groups with 173 (7.7%) to 239 (10.7%) children each. The majority of households were Muslim (86.6%), and half of mothers (50.2%) and fathers (49.0%) had less than a primary education. The highest household asset-based wealth quintile, calculated based on the entire MDSS, comprised 32.9% of the sample and the lowest comprised 5.6%.

**Table 2. 12-month prevalence of atopic dermatitis according to the International Study of Asthma and Allergies in Childhood (ISAAC) questionnaire [a].**

| Age[b] | N | n | % | Lower confidence interval | Upper confidence interval |
|---|---|---|---|---|---|
| < 6 months | 235 | 19 | 8.1% | 4.6% | 11.6% |
| 6–11 months | 215 | 24 | 11.2% | 7.0% | 15.4% |
| 12–17 months | 239 | 29 | 12.1% | 8.0% | 16.3% |
| 18–23 months | 234 | 33 | 14.1% | 9.6% | 18.6% |
| 24–29 months | 230 | 31 | 13.5% | 9.1% | 17.9% |
| 30–35 months | 228 | 37 | 16.2% | 11.4% | 21.0% |
| 36–41 months | 229 | 32 | 14.0% | 9.5% | 18.5% |
| 42–47 months | 217 | 24 | 11.1% | 6.9% | 15.2% |
| 48–53 months | 242 | 20 | 8.3% | 4.8% | 11.7% |
| 54–59 months | 173 | 18 | 10.4% | 5.9% | 15.0% |
| Overall | 2242 | 267 | 11.9% | 10.6% | 13.3% |

[a]Atopic dermatitis was defined as an itchy rash at any time coming and going for at least 6 months, that had at any time affected the folds of the elbows, behind the knees, in front of the ankles, under the buttocks, or around the neck, ears or eyes in the past 12 months.

[b]For children <6 months of age, a positive screen was determined by an affirmative response to the ISAAC question regarding the anatomical location of the rash.

## Prevalence of atopic dermatitis

ISAAC identified 267 children with AD: an overall prevalence of 11.9% [95% confidence interval (CI) 10.6–13.3] (Table 2). The prevalence increased from 8.1% (95% CI 4.6–11.6) in children <6 months to a high of 16.2% (95% CI 11.4–21.0) in children 30–35 months and decreased to 8.3% (95% CI 4.8–11.7) in children 48–53 months. Severe disease was identified in 11.7% of all cases. There was no difference in prevalence by sex (p = 0.074), levels of maternal education (p = 0.457), or asset score (p = 0.571). Only 10 children (3.7%)– 9 males, 1 female–had ever received a physician's diagnosis of AD.

Of the 266 participants identified with AD and for whom we had POEM scores, there were 144 (54.1%) clear, 52 (19.6%) mild, 60 (22.6%) moderate, and 10 (3.8%) severe cases. POEM did not significantly differ between 6-month age groups (p = 0.061). The component questions of the POEM score as they relate to all cases and ISAAC severe cases are shown in Table 3. Chi-squared analysis of the POEM categories and ISAAC severe cases was borderline significant (p = 0.051) (Table 4). Of caretakers who reported that their child was awakened once or more weekly on average over the past 12 months for ISAAC, 48.4% responded that their child's

**Table 3. Components of POEM severity measure[a].**

| | All cases, N = 266 % (n) | | | | | ISAAC severe cases, N = 31 % (n) | | | | |
|---|---|---|---|---|---|---|---|---|---|---|
| | No days | 1–2 days | 3–4 days | 5–6 days | Everyday | No days | 1–2 days | 3–4 days | 5–6 days | Everyday |
| Skin been itchy? | 50.9 (135) | 13.2 (35) | 3.8 (10) | 3.4 (9) | 28.7 (76) | 29.0 (9) | 12.9 (4) | 3.2 (1) | 3.2 (1) | 51.6 (16) |
| Sleep been disturbed? | 80.8 (215) | 8.3 (22) | 3.8 (10) | 1.1 (3) | 6.0 (16) | 48.4 (15) | 16.1 (5) | 9.7 (3) | 0.0 (0) | 25.8 (8) |
| Skin been bleeding? | 82.3 (218) | 7.2 (19) | 4.5 (12) | 1.9 (5) | 4.2 (11) | 71.0 (22) | 12.9 (4) | 3.2 (1) | 3.2 (1) | 6.5 (2) |
| Skin been weeping? | 94.0 (250) | 3.0 (8) | 2.3 (6) | 0.0 (0) | 0.8 (2) | 93.6 (29) | 3.2 (1) | 0.0 (0) | 0.0 (0) | 3.2 (1) |
| Skin been cracked? | 89.4 (237) | 3.8 (10) | 2.3 (6) | 1.1 (3) | 3.4 (9) | 77.4 (24) | 12.9 (4) | 3.2 (1) | 3.2 (1) | 3.2 (1) |
| Skin been flaking off? | 66.0 (175) | 14.3 (38) | 6.4 (17) | 3.4 (9) | 9.8 (26) | 54.8 (17) | 25.8 (8) | 3.2 (1) | 3.2 (1) | 12.9 (4) |
| Skin felt rough or dry? | 64.3 (171) | 15.4 (41) | 3.0 (8) | 2.3 (6) | 15.0 (40) | 51.6 (16) | 22.6 (7) | 6.5 (2) | 0.0 (0) | 19.4 (6) |

[a] The POEM question stem is, "Over the past week, on how many nights has your child's <see first column> because of their eczema?".

**Table 4. Concordance between ISAAC and POEM severity measures.**

| ISAAC | POEM | | | |
|---|---|---|---|---|
| | **Clear** | **Mild** | **Moderate** | **Severe** |
| **Not severe** | 57.1 (133) | 18.5 (43) | 21.0 (49) | 3.4 (8) |
| **Severe** | 29.0 (9) | 29.0 (9) | 35.5 (11) | 6.5 (2) |

sleep had been disturbed 0 days, 16.1% 1–2 days, 9.7% 3–4 days, 0.0% 5–6 days, and 25.8% every day.

## Psychosocial measures

**Infants' Dermatitis Quality of Life.** The mean IDQoL score was 2.32 (SD 3.2, range 1–18) and ranged non-significantly from 2.00 (SD 3.26) to 2.83 (SD 3.07) between the ten 6-month age groups (p = 0.933) (Table 5). Males (2.67, SD 3.42) had significantly higher IDQoL scores than females (1.95, SD 2.91, p = 0.015). Mean IDQoL scores by ISAAC severity categories were 1.83 (SD 2.70) for no AD, 2.09 (SD 2.87) for non-severe AD, and 5.19 (SD 4.34) for severe AD (p<0.001). There were significant differences in IDQoL scores between POEM categories: clear 0.90 (SD 2.10), mild 3.20 (SD 2.60), moderate 4.80 (SD 3.30), and severe 9.40 (SD 3.60) (p<0.001). IDQoL scores were also significantly different between the lowest (3.06, SD 3.60) and highest wealth quintiles (1.63, SD 2.72; p<0.001). IDQoL was higher among mothers with <primary (2.41, SD 2.84) and <secondary education (2.47, SD 3.63) compared to those with >secondary education (1.43, SD 2.19; p = 0.039).

Caretakers rated the severity of their child's AD as extremely severe (9.4%), severe (10.9%), average (0.4%), fairly good (30.8%), or none (48.5%). There was no significant difference in caretakers' rating of disease as severe between male (21.5%) and female children (18.6%) (p = 0.587). Children perceived by their caretakers to have severe AD had higher IDQoL scores compared to those not rated as severe by caretakers, (5.83, SD 3.40 vs 1.46, SD 2.26; p<0.001).

Among all children with AD, the greatest contribution to IDQoL scores was itching and scratching: 11.3% reported it to be present all of the time and 11.7% as a lot of the time (Table 6). Mood was the next largest contributor: 1.9% reported their child as extremely irritable/always crying and 8.3% as very irritable. More than 90% of caretakers reported that AD did not interfere with playing or the child taking part in family activities or mealtimes, nor were there problems caused by treatment. When restricted to only participants whose caretakers rated their AD as severe or extremely severe, itching and scratching was again the largest contributor to IDQoL scores– 38.9% reported itching and scratching all of the time, and 27.8% a lot of the time–and mood was second, with 9.3% of children described as extremely irritable/always crying and 35.2% described as very irritable.

**Dermatitis family impact.** Overall mean DFI score was 1.18 (SD 2.85, range 0–22), ranging from 0.52 (SD 2.18) to 1.38 (SD 3.90, p = 0.937) among the 10 age groups (Table 5). Males (1.36) and females (0.98) had similar mean DFI scores (p = 0.057). DFI scores differed significantly by ISAAC categories: negative 0.84 (SD 1.70), non-severe 0.97 (SD 2.40), and severe 3.32 (SD 5.46, p<0.001). DFI was also significantly different between POEM categories: clear 0.29 (SD 1.09), mild 1.42 (SD 2.77), moderate 2.60 (SD 3.85), and severe 8.55 (SD 5.54, p<0.001). DFI was significantly higher for children whose AD was rated as severe by caretakers compared to those with AD not rated as severe (9.71, SD 7.78 vs. 0.57, SD 1.62; p<0.001). There was no significant difference in DFI between wealth quintiles (p = 0.079) or maternal education (p = 0.131).

**Table 5. Infants' Dermatitis Quality of Life (IDQoL) and Dermatitis Family Impact (DFI) scores.**

| | IDQoL | | | DFI | | |
|---|---|---|---|---|---|---|
| | Mean | Standard deviation | p-value | Mean | Standard deviation | p-value |
| **Overall** | 2.32 | 3.20 | – – | 1.18 | 2.85 | – – |
| **Age group** | | | | | | |
| < 6 months | 2.08 | 2.33 | 0.933 | 1.20 | 2.38 | 0.937 |
| 6–11 months | 2.00 | 3.26 | | 1.32 | 3.90 | |
| 12–17 months | 2.16 | 3.35 | | 1.13 | 3.52 | |
| 18–23 months | 2.74 | 4.22 | | 1.30 | 2.62 | |
| 24–29 months | 2.02 | 2.93 | | 1.29 | 3.11 | |
| 30–35 months | 2.15 | 2.72 | | 1.21 | 2.79 | |
| 36–41 months | 2.83 | 3.07 | | 1.17 | 2.24 | |
| 42–47 months | 2.18 | 3.13 | | 0.64 | 1.29 | |
| 48–53 months | 2.53 | 3.30 | | 1.69 | 3.29 | |
| 54–59 months | 2.33 | 3.45 | | 0.52 | 2.18 | |
| **Sex** | | | | | | |
| Male | 2.67 | 3.42 | 0.023 | 1.38 | 3.06 | 0.164 |
| Female | 1.95 | 2.91 | | 0.98 | 2.60 | |
| **ISAAC[a]** | | | | | | |
| Negative | 1.83 | 2.70 | <0.001 | 0.84 | 1.70 | <0.001 |
| Positive, not severe | 2.09 | 2.87 | | 0.97 | 2.40 | |
| Positive, severe | 5.19 | 4.34 | | 3.32 | 5.46 | |
| **POEM[b]** | | | | | | |
| Clear | 0.90 | 2.10 | <0.001 | 0.29 | 1.09 | <0.001 |
| Mild | 3.20 | 2.60 | | 1.42 | 2.77 | |
| Moderate | 4.80 | 3.30 | | 2.60 | 3.85 | |
| Severe | 9.40 | 3.60 | | 8.55 | 5.54 | |
| **Caregiver rated severity** | | | | | | |
| Not severe | 1.46 | 2.26 | <0.001 | 0.57 | 1.62 | <0.001 |
| Severe | 5.83 | 3.40 | | 9.71 | 7.78 | |
| **Wealth quintile** | | | | | | |
| Low | 3.06 | 3.60 | 0.002 | 0.82 | 1.70 | 0.304 |
| Middle low | 2.32 | 2.88 | | 1.34 | 2.32 | |
| Middle | 3.34 | 3.44 | | 1.65 | 3.58 | |
| Middle high | 2.19 | 3.54 | | 1.09 | 3.12 | |
| High | 1.63 | 2.72 | | 0.86 | 2.40 | |
| **Maternal education** | | | | | | |
| Less than primary | 2.41 | 2.84 | 0.078 | 1.12 | 2.35 | 0.512 |
| Some secondary | 2.47 | 3.63 | | 1.30 | 3.20 | |
| Secondary or higher | 1.43 | 2.19 | | 0.80 | 2.43 | |

[a]ISAAC = International Study of Asthma and Allergies in Childhood. Atopic dermatitis (AD) was defined as an itchy rash at any time coming and going for at least 6 months, that had at any time affected the folds of the elbows, behind the knees, in front of the ankles, under the buttocks, or around the neck, ears or eyes in the past 12 months. Severe AD was defined as being kept awake one or more nights per week on average by this itchy rash in the past 12 months.

[b]POEM = Patient-Oriented Eczema Measure.

Eighty-six percent or more of caretakers responded "Not at all" to each DFI question. When restricted to those with AD reported as severe, 16.7% said that their child's AD caused them "A lot" or "Very much" emotional distress. Housework was the most affected activity (13.0%) (Table 6). Food preparation, sleep of family members, and

**Table 6. Components of quality of life measures.**

| | All cases, N = 266% (n) | | | | Severe cases, N = 54[a] % (n) | | | |
|---|---|---|---|---|---|---|---|---|
| **Infants' Dermatitis Quality of Life** | **0** | **1** | **2** | **3** | **0** | **1** | **2** | **3** |
| How much has your child been itching and scratching? | 47.0% (125) | 30.1% (80) | 11.7% (31) | 11.3% (30) | 7.4% (4) | 25.9% (14) | 27.8% (15) | 38.9% (21) |
| What has your child's mood been? | 64.5% (171) | 25.3% (67) | 8.3% (22) | 1.9% (5) | 25.9% (14) | 29.6% (16) | 35.2% (19) | 9.3% (5) |
| Approximately how much time on average has it taken to get your child off to sleep each night? | 74.7% (198) | 22.3% (59) | 3.0% (8) | 0.0% (0) | 58.6% (30) | 35.9% (19) | 7.6% (4) | 0.0% (0) |
| What was the total time that your child's sleep was disturbed on average each night? | 85.7% (288) | 12.0% (32) | 1.9% (5) | 0.4% (1) | 59.3% (32) | 35.2% (19) | 5.6% (3) | 0.0% (0) |
| Has your child's eczema interfered with playing? | 91.7% (243) | 7.2% (19) | 0.8% (2) | 0.4% (1) | 74.1% (40) | 20.4% (11) | 3.7% (2) | 1.9% (1) |
| Has your child's eczema interfered with your child taking part in or enjoying other family activities? | 94.4% (251) | 5.3% (14) | 0.4% (1) | 0.0% (0) | 83.3% (45) | 14.8% (8) | 1.9% (1) | 0.0% (0) |
| Have there been problems with your child at mealtimes because of the eczema? | 90.3% (240) | 7.9% (21) | 1.5% (4) | 0.4% (1) | 75.9% (41) | 16.7% (9) | 5.6% (3) | 1.9% (1) |
| Have there been problems with your child caused by the treatment? | 92.5% (245) | 6.4% (17) | 1.1% (3) | 0.0% (0) | 81.5% (44) | 13.0% (7) | 5.6% (3) | 0.0% (0) |
| Has your child's eczema meant that dressing and undressing the child has been uncomfortable? | 89.9% (239) | 7.5% (20) | 2.6% (7) | 0.0% (0) | 75.9% (41) | 11.1% (6) | 13.0% (7) | 0.0% (0) |
| How much has your child having eczema been a problem at bath time? | 88.4% (235) | 8.7% (23) | 3.0% (8) | 0.0% (0) | 72.2% (39) | 18.5% (10) | 9.3% (5) | 0.0% (0) |
| **Dermatitis Family Index** | | | | | | | | |
| Housework, e.g. washing, cleaning? | 86.8% (231) | 10.2% (27) | 2.6% (7) | 0.4% (1) | 68.5% (37) | 18.5% (10) | 11.1% (6) | 1.9% (1) |
| Food preparation and feeding? | 89.9% (239) | 7.5% (20) | 1.3% (4) | 1.1% (3) | 70.4% (38) | 18.5% (10) | 5.6% (3) | 5.6% (3) |
| The sleep of other in the family? | 88.0% (234) | 9.8% (26) | 1.5% (4) | 0.8% (2) | 64.8% (35) | 24.1% (13) | 7.4% (4) | 3.7% (2) |
| Family leisure activities? | 91.4% (243) | 6.8% (18) | 1.9% (5) | 0.0% (0) | 79.6% (43) | 11.1% (6) | 9.3% (5) | 0.0% (0) |
| Time spent going to the market? | 93.6% (249) | 6.0 (16) | 0.4% (1) | 0.0% (0) | 79.6% (43) | 18.5% (10) | 1.9% (1) | 0.0% (0) |
| Your expenditures, e.g. costs related to treatment, clothes, etc.? | 91.7% (244) | 6.0% (16) | 1.9% (5) | 0.4% (1) | 74.1% (40) | 14.8% (8) | 9.3% (5) | 1.9% (1) |
| On causing tiredness or exhaustion in you or your husband or other family members who care for your child? | 92.1% (245) | 6.0% (16) | 1.1% (3) | 0.8% (2) | 74.1% (40) | 16.7% (9) | 5.6% (3) | 3.7% (2) |
| Causing you emotional distress such as depression, frustration, or guilt? | 87.9% (233) | 8.3% (22) | 2.3% (6) | 1.5% (4) | 68.5% (37) | 14.8% (8) | 11.1% (6) | 5.6% (3) |
| Relationships between you and your husband or you and your other children? | 94.7% (252) | 3.8% (10) | 1.1% (3) | 0.4% (1) | 81.5% (44) | 11.1% (6) | 5.6% (3) | 1.9% (1) |
| Helping with your child's treatment had on your life? | 87.0% (227) | 10.7% (28) | 1.5% (4) | 0.8% (2) | 52.8% (28) | 35.9% (19) | 7.6% (4) | 3.8% (2) |

[a] As defined by caretakers rating their child's atopic dermatitis as "extremely severe" or "severe."

expenditures were also affected (11.2% each). Time spent going to the market was the least affected (1.9%).

The IDQoL and DFI had a strong positive correlation (r = 0.66 p<0.001). Similarly, POEM, taken as a continuous variable, had strong positive correlations with both IDQoL (r = 0.77, p<0.001) and DFI (r = 0.56, p<0.001). The caregiver's rating of severe disease was also strongly positively correlated with POEM (r = 0.73, p<0.001), IDQoL (r = 0.82, p<0.001) and DFI (r = 0.57, p<0.001).

## Discussion

This is the first study conducted in Bangladesh that reports both the prevalence of AD in children under 5 and its associated psychosocial impacts on affected children and their families. AD affected more than 1 in 10 children. Similar to studies performed in HICs, the highest prevalence of AD (16%) was found in children at 2 years of age [34–36]. Severe disease was found in 3.8% by POEM and 11.7% by ISAAC. A 2005 study in Bangladesh reported the prevalence of atopic dermatitis to be 6.0% and 7.1% in children 6–7 and 13–14 years, respectively [16]. However, previous estimates for AD prevalence in children under 5 and data on AD severity in Bangladesh are lacking. The ISAAC Phase 3 trials, which did not include Bangladesh, and additional studies in China [37] and Japan [38], found that approximately 12% of children aged 6–7 and 13–14 in the Asia-Pacific region had severe disease [2], similar to the ISAAC-measured severity reported here. While the patterns are the same, AD prevalence is lower than that reported in HICs. This was also seen in Indian centers, which were among those with the lowest AD prevalence globally in the ISAAC trials. Notably, among these centers, there were large variations in prevalence, between 0.9–9.2% [39]. Variations in prevalence, both globally and within countries, have yet to be explained. AD is known to be multifactorial with genetics and environmental exposures playing important etiologic roles. The hygiene hypothesis, a widely accepted explanation for the higher prevalence of immunoregulatory disorders in HICs compared to LMICs, suggests that reduced exposure to microbes early in life, such as through better housing, and access to health care, detergents and running water, alters the body's natural microbiome that helps protect against these disorders, including atopic dermatitis. As Bangladesh continues to see improvements in these sectors, prevalence of such conditions may continue to increase though it is not as straightforward as it once seemed; data regarding seemingly similar exposures is heterogeneous. Childhood helminth exposures, for example, have shown both protective and harmful effects depending on the species [40,41].

Notably, only 3.7% of children with AD had ever received formal diagnosis by a physician. This reflects the poor access to healthcare in this population. While the ISAAC questionnaire has a standard question for doctor's diagnosis, it is not routinely reported in trial results. Data from HICs was not available; however, a study of 1,704 children in Nigeria reported an AD prevalence of 10.1% and that 8.7% had received a formal diagnosis [42].

Children and families of children with severe disease were significantly more likely to experience reduced quality of life as a consequence of AD. Severity measured by ISAAC was higher than that measured by POEM likely because of the temporal aspect and the number of questions for each measure. ISAAC requires only that the child have sleep disturbed by AD symptoms once or more weekly on average in the year prior whereas POEM asks about multiple symptoms in the past week and requires more affirmative responses to be scored as severe. The inconsistency in severity ratings could also be due to the seasonal nature of AD and its greater potential influence on POEM scoring. The component questions of POEM among all cases and ISAAC severe cases are described in Table 3.

Caretakers were more likely to rate their child's AD as severe compared to ISAAC and POEM measures. Over-estimation by caregivers has been previously documented [43]. However, as found in multiple other studies, caregiver-rated severity was positively correlated with both quality of life measures suggesting that their severity assessment might be more holistic, taking into account both physical symptoms and the psychosocial aspects of the disease [27,29]. In this study, consistent with others [43,44], caretakers who perceived their child to have severe disease reported more emotional distress and disruptions to various aspects of daily life.

Although there was no significant difference in IDQoL or DFI overall between those with and without AD, both IDQoL and DFI increased in a step-wise manner with severity of disease as measured by ISAAC, POEM, and caretakers' perception. Previous studies have similarly shown no reduction in quality of life in children with mild AD but correlation between increasing severity and IDQoL [27,30]. Similar to Italian [43] and U.K. studies [30], itching and scratching and mood were perceived to be the symptoms that most negatively affected children, especially those with severe disease.

Males had significantly higher IDQoL scores and borderline significantly higher DFI compared to females. Reasons for the result are unclear, however, a Singapore study found that males under 4 years had higher IDQoL scores compared to females of the same age, although the trend reversed at age 5 likely due to AD having a greater social impact on females [45]. A study in Ukraine among children <4 years found lower quality of life among females [46]. These findings suggest that sex-specific impacts are contextual, and support for patients and their families may be beneficial.

Low wealth and maternal education both were associated with higher IDQoL scores but not with DFI. The higher IDQoL scores could be due to limited access to treatments and management of the disease, whereas the magnitude of other stressors in the environment could influence the perceived lack of effect on DFI. Further research is needed to explore sex-specific effects of AD on quality of life–specifically the perceived higher impacts of AD on quality of life for males in Bangladesh–and the lack of effects of the child's disease on poorer, less educated families.

IDQoL and DFI have been correlated in multiple studies [27]. In our study, severity as measured by POEM was more strongly correlated with IDQoL than DFI. This correlation is likely explained by IDQoL and POEM both scoring symptoms and specifically the frequency of itching and scratching (the most common symptom reported), which is the largest contributor to IDQoL. In contrast, DFI does not include questions about symptoms experienced by the child, and focuses instead on reduction in activities and the family's well-being due to the child's AD.

## Limitations

While our sample was adequately powered to calculate the prevalence of AD, it should be noted that variation in AD prevalence and severity across the country is possible due to multiple factors including genetic and environmental variations [17]. Secondly, our research [15] has demonstrated that for children <1 year in this population, AD prevalence as determined by ISAAC matched that of the U.K. Criteria, which has been validated in this age group. However, ISAAC prevalence was lower for children 1–4 years compared to U.K. Criteria, suggesting that our results may be an underestimation of AD prevalence in children 1–4 years. Additionally, while ISAAC is a validated methodology that has been used internationally to measure the prevalence of AD in various races and ethnicities, it has not been validated in Bangladesh. Research from Ethiopia and South Africa that aimed to validate ISAAC and the U.K. Criteria against a physician's diagnosis suggests that these tools may have low sensitivity in non-Caucasian populations [47,48]. Lastly, the scores reported here for both IDQoL and DFI were lower than those typically reported in HICs. IDQoL and DFI scores tend to correlate together [29,30] but vary greatly from study to study likely due to selection criteria, study setting, cultural modifications, and translations made to the questionnaire; comparisons of scores by the two methods are rarely made for these reasons [27]. Although all questionnaires were translated using the ISAAC Phase 3 Manual guidelines, it is possible that the translations missed important nuances resulting in caretakers ranking those characteristics lower than they typically would. For example, we modified the DFI question regarding "time spent shopping

for the family" to the more culturally appropriate phrase, "time spent going to the market," yet this was the DFI question that had the fewest affirmative responses. In Bangladeshi homes, homemakers typically visit the market daily so it is possible that this crucial daily event is less likely to be affected by a child's AD. Stronger social ties and family structures in LMICs compared to HICs may also have the effect of reducing the burden of AD on family members [49]. It is also possible that perceived impacts of AD are altered in an environment with multiple stressors [50].

## Conclusion

As the prevalence of AD increases in LMICs, it is crucial to understand not only the prevalence and pattern of disease but also its psychosocial implications. Here, we show that AD, especially severe AD, is associated with significant psychosocial morbidity among both children and their families. As access to healthcare expands in LMICs, it is essential that we can identify infants at risk for severe disease and provide education, support, and treatment that addresses both the medical and psychosocial aspects of disease management for the child and their family.

## Supporting information

**S1 Checklist. STROBE statement—Checklist of items that should be included in reports of cross-sectional studies.**
(DOC)

## Author Contributions

**Conceptualization:** Courtney J. Pedersen, Gary L. Darmstadt.

**Data curation:** Courtney J. Pedersen, Mohammad J. Uddin, Samir K. Saha.

**Formal analysis:** Courtney J. Pedersen.

**Supervision:** Gary L. Darmstadt.

**Writing – original draft:** Courtney J. Pedersen, Gary L. Darmstadt.

**Writing – review & editing:** Courtney J. Pedersen, Mohammad J. Uddin, Samir K. Saha, Gary L. Darmstadt.

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
