## [Decision Letter · Decision Letter 0]

12 Mar 2021

PONE-D-20-34512

Prevalence and psychosocial impact of atopic dermatitis in Bangladeshi children and families

PLOS ONE

Dear Dr. Darmstadt,

Thank you for submitting your manuscript to PLOS ONE. After careful consideration, we feel that it has merit but does not fully meet PLOS ONE’s publication criteria as it currently stands. Therefore, we invite you to submit a revised version of the manuscript that addresses the points raised during the review process.

We look forward to receiving your revised manuscript.

Kind regards,

Dong Keon Yon, MD, FACAAI

Academic Editor

PLOS ONE

Journal Requirements:

3. Please include additional information regarding the survey or questionnaire used in the study and ensure that you have provided sufficient details that others could replicate the analyses.

For instance, if you developed a questionnaire as part of this study and it is not under a copyright more restrictive than CC-BY, please include a copy, in both the original language and English, as Supporting Information, or include a citation if it has been published previously.

4. In your statistical analyses, please state whether you accounted for clustering by region or village.

For example, did you consider using multilevel models?

5. Please include captions for your Supporting Information files at the end of your manuscript, and update any in-text citations to match accordingly. Please see our Supporting Information guidelines for more information: http://journals.plos.org/plosone/s/supporting-information

Additional Editor Comments:

Thank you for submitting your perfect manuscript to Plos One. The reviewers and I believe it is of potential value for our readers. However, the reviewers have raised a number of very important issues, and their excellent comments will need to be adequately addressed in a revision before the acceptability of your manuscript for publication in the Journal can be determined.

Editor major comments: None

Reviewers' comments:

Reviewer's Responses to Questions

**Comments to the Author**

1. Is the manuscript technically sound, and do the data support the conclusions?

Reviewer #1: Yes

Reviewer #2: Yes

Reviewer #3: Yes

Reviewer #4: Yes

2. Has the statistical analysis been performed appropriately and rigorously? 

Reviewer #1: Yes

Reviewer #2: Yes

Reviewer #3: Yes

Reviewer #4: Yes

3. Have the authors made all data underlying the findings in their manuscript fully available?

Reviewer #1: Yes

Reviewer #2: Yes

Reviewer #3: Yes

Reviewer #4: Yes

4. Is the manuscript presented in an intelligible fashion and written in standard English?

Reviewer #1: Yes

Reviewer #2: Yes

Reviewer #3: Yes

Reviewer #4: Yes

5. Review Comments to the Author

Reviewer #1: This is a nice study evaluating the burden of atopic dermatitis in under-5 children in the district of Bangladesh. Overall the methodology is valid including the use of standardized tools and the results are clear. I have no particular points. Albeit descriptive, the study succeds in underscoring the prevalence of AD, its impact on QoL and associations with certain sociodemographic characteristics.

Reviewer #2: Pedersen et al present an important evaluation on the prevalence and psychosocial impact of AD in Bangali children and families. This manuscript provides important information based on cross-sectional surveys regarding how the prevalence of an atopic condition, such as AD, can vary especially when comparing to the prevalence rates to different regions of the world that are more industrialized and have different wealth and educational outcomes.

Overall, the paper is clear and concise. Points of discussion:

1. As the authors have pointed out the ISAAC and POEM both measured the severity of the disease. Severity measured by ISAAC was HIGHER than that measured by POEM, which as the authors state was likely related to the temporal aspect and number of questions for each measure. Pointing out these differences is important to the reader who may want to choose using one survey over the other and how the results can impact the overall data if the survey, such as POEM, is utilized at a time of year when AD is controlled. I would consider including a table illustrating the different component questions to these 2 surveys for the reviewers to understand and appreciate given their variability of assessment. Consider a Table layout similar to the one you have for Table 4.

2. Interestingly, the prevalence of disease severity in this cohort was just as high as the ISAAC Phase 3 trials, the Chinese and Japanese studies. However, the AD prevalence was lower than that in HICs. In addition to the statement "AD is known to be multifactorial with genetics and environmental exposures playing important etiologic roles." I would more specifically include how differences in industrialization and the potential to health care access can effect these patterns of prevalence which may explain the lower prevalences in India and Bangladesh.

3. The statement made just before table 2 "Only 10 children - 9 males, 1 female- have ever received a physician's diagnosis of AD" is important to emphasize to the readers and how this differs between other countries that report prevalence of AD with higher rates of physician diagnosed AD.

Minor points:

1. I would include Line numbers for future manuscript publications to make it easier to reference areas of your manuscript

2. Table 3 - in the DFI section, align the p-values to all be in the same alignment

Reviewer #3: The study reports interesting data on AD in Bangladesh. It has a large sample size, which in a rural setting may be a real challenge. The authors present a thorough investigation of impact of AD on quality of life in Bangladesh.

Minor comments to the authors

1. In the introduction, authors state that AD prevalence is increasing in low and middle income countries. There is the possibility of actual increase of AD and /or an increase in access to the diagnosis of AD. It would be hard to know whether prevalence is really increasing or health care is now more available and thus cases are being identified. The authors should consider making a statement about this.

2. Discussion: Since children where not evaluated by a health care worker, can we say that the study reports prevalence? Could authors give their view on the ability of ISAAC questionnaire to estimate prevalence? Has the score validated the diagnosis of AD through comparison with actual evaluations by health care workers?

Reviewer #4: DEAR EDITOR, I REVIEWED THE ARTICLE ENTITLED AS "Prevalence and psychosocial impact of atopic dermatitis in Bangladeshi children and families" FOR PLOS ONE. IT WAS A WELL WRITTEN ARTICLE DESCRIBING THE EPIDEMIOLOGIC CHARACTERISTICS OF ATOPIC DERMATITIS IN A LOCAL REGION. IT WOULD BE VALUABLE TO PUBLISH THIS LOCAL INFORMATION TO THE WHOLE WORLD.

KIND REGARDS

6. PLOS authors have the option to publish the peer review history of their article (what does this mean?). If published, this will include your full peer review and any attached files.

Reviewer #1: No

Reviewer #2: No

Reviewer #3: No

Reviewer #4: **Yes: **AYSE BACCIOGLU

---

## [Author Response · Author response to Decision Letter 0]

23 Mar 2021

Dear Dr. Yon, 

Thank you for your interest in our manuscript (PONE-D-20-34512) entitled “Prevalence and psychosocial impact of atopic dermatitis in Bangladeshi children and families.”

We thank all the Reviewers and the Editor for their thoughtful and detailed comments. Below please find our point-by-point responses to these comments. We have numbered and bolded each comment, and our responses follow in plain text. Corresponding changes in the manuscript have been made using track changes. As requested, our resubmission includes one copy of the manuscript with changes tracked and one “clean” copy of the manuscript with all changes accepted. Please note that the line references in our responses below refer to the clean version of the revised manuscript. 

Editor

We have double-checked the references. They are complete and correct. We’ve added five references (40, 41, 42, 47, and 48) in the Discussion in response to the Reviewer’s suggested edits.

We have double-checked the formatting and made edits as indicated. 

3. Please include additional information regarding the survey or questionnaire used in the study and ensure that you have provided sufficient details that others could replicate the analyses.

For instance, if you developed a questionnaire as part of this study and it is not under a copyright more restrictive than CC-BY, please include a copy, in both the original language and English, as Supporting Information, or include a citation if it has been published previously.

We used publicly available questionnaires. However, we did make two minor changes, which are summarized in the Methods on lines 119-124 as follows:

“All questionnaires were translated from English to Bangla by study physicians, pilot tested in the community, and adjusted after discussions with CHWs per ISAAC Phase 3 Manual recommendations [31]. Question 6 and question 4 in the IDQoL and DFI, respectively, were modified by removing the example of swimming as it was not culturally relevant. Question 5 in the DFI was modified from asking about “time spent on shopping for the family” to “time spent going to the market.”

4. In your statistical analyses, please state whether you accounted for clustering by region or village.

For example, did you consider using multilevel models?

The study population comprised of a random sample of clusters that are balanced to the Mirzapur Demographic Surveillance Site population. Thus, weighing by region or village was not taken into account in the analysis. We have updated our Methods section in lines 61 – 63 and lines 130 – 132:

“All 156 villages in the MDSS were subdivided into 110 clusters each and ten clusters were randomly chosen to be included in the study, approximating a population-based sample.”

“AD prevalence was calculated using the total number of individuals in each age group for the denominator for that age group; clustering by region or village was not taken into account.”

5. Please include captions for your Supporting Information files at the end of your manuscript, and update any in-text citations to match accordingly. Please see our Supporting Information guidelines for more information: http://journals.plos.org/plosone/s/supporting-information

No supporting materials are included.

Reviewer 1

This is a nice study evaluating the burden of atopic dermatitis in under-5 children in the district of Bangladesh. Overall the methodology is valid including the use of standardized tools and the results are clear. I have no particular points. Albeit descriptive, the study succeeds in underscoring the prevalence of AD, its impact on QoL and associations with certain sociodemographic characteristics.

We thank Reviewer 1 for their review of our manuscript and these helpful comments. 

Reviewer 2

Pedersen et al present an important evaluation on the prevalence and psychosocial impact of AD in Bengali children and families. This manuscript provides important information based on cross-sectional surveys regarding how the prevalence of an atopic condition, such as AD, can vary especially when comparing to the prevalence rates to different regions of the world that are more industrialized and have different wealth and educational outcomes.

Overall, the paper is clear and concise. Points of discussion:

1. As the authors have pointed out the ISAAC and POEM both measured the severity of the disease. Severity measured by ISAAC was HIGHER than that measured by POEM, which as the authors state was likely related to the temporal aspect and number of questions for each measure. Pointing out these differences is important to the reader who may want to choose using one survey over the other and how the results can impact the overall data if the survey, such as POEM, is utilized at a time of year when AD is controlled. I would consider including a table illustrating the different component questions to these 2 surveys for the reviewers to understand and appreciate given their variability of assessment. Consider a Table layout similar to the one you have for Table 4.

We agree that choosing between these two methods of measuring severity is an important consideration when finalizing study protocols. As demonstrated by the Harmonising Outcome Measures for Eczema working group, there is international interest in standardizing research protocols for AD such that studies can be compared across populations. While our study was not designed to compare and validate these measures against one another, your point is well taken that future researchers may benefit from additional information on the component questions. We have created two tables additional for our manuscript that break down these components for the reader. Per your helpful recommendation, our new Table 3 is similar to the layout of the original Table 4, which is now labeled Table 6.

We have pointed interested researchers to these tables in lines 178-186 and lines 299-308, respectively, as follows:

“Of the 266 participants identified with AD and for whom we had POEM scores, there were 144 (54.1%) clear, 52 (19.6%) mild, 60 (22.6%) moderate, and 10 (3.8%) severe cases. POEM did not significantly differ between 6-month age groups (p=0.061). The component questions of the POEM score as they relate to all cases and ISAAC severe cases are shown in Table 3. Chi-squared analysis of the POEM categories and ISAAC severe cases was borderline significant (p=0.051) (Table 4). Of caretakers who reported that their child was awakened once or more weekly on average over the past 12 months for ISAAC, 48.4% responded that their child’s sleep had been disturbed 0 days, 16.1% 1-2 days, 9.7% 3-4 days, 0.0% 5-6 days, and 25.8% every day.”

“Children and families of children with severe disease were significantly more likely to experience reduced quality of life as a consequence of AD. Severity measured by ISAAC was higher than that measured by POEM likely because of the temporal aspect and the number of questions for each measure. ISAAC requires only that the child have sleep disturbed by AD symptoms once or more weekly on average in the year prior whereas POEM asks about multiple symptoms in the past week and requires more affirmative responses to be scored as severe. The inconsistency in severity ratings could also be due to the seasonal nature of AD and its greater potential influence on POEM scoring. The component questions of POEM among all cases and ISAAC severe cases are described in Table 3.”

Table 3. Components of POEM severity measurea

 All cases, N = 266

% (n) ISAAC severe cases, N = 31

% (n)

 No days 1-2 days 3-4 days 5-6 days Everyday No days 1-2 days 3-4 days 5-6 days Everyday

Skin been itchy? 50.9 (135) 13.2 (35) 3.8 (10) 3.4 (9) 28.7 (76) 29.0 (9) 12.9 (4) 3.2 (1) 3.2 (1) 51.6 (16)

Sleep been disturbed? 80.8 (215) 8.3 (22) 3.8 (10) 1.1 (3) 6.0 (16) 48.4 (15) 16.1 (5) 9.7 (3) 0.0 (0) 25.8 (8)

Skin been bleeding? 82.3 (218) 7.2 (19) 4.5 (12) 1.9 (5) 4.2 (11) 71.0 (22) 12.9 (4) 3.2 (1) 3.2 (1) 6.5 (2)

Skin been weeping? 94.0 (250) 3.0 (8) 2.3 (6) 0.0 (0) 0.8 (2) 93.6 (29) 3.2 (1) 0.0 (0) 0.0 (0) 3.2 (1)

Skin been cracked? 89.4 (237) 3.8 (10) 2.3 (6) 1.1 (3) 3.4 (9) 77.4 (24) 12.9 (4) 3.2 (1) 3.2 (1) 3.2 (1)

Skin been flaking off? 66.0 (175) 14.3 (38) 6.4 (17) 3.4 (9) 9.8 (26) 54.8 (17) 25.8 (8) 3.2 (1) 3.2 (1) 12.9 (4)

Skin felt rough or dry? 64.3 (171) 15.4 (41) 3.0 (8) 2.3 (6) 15.0 (40) 51.6 (16) 22.6 (7) 6.5 (2) 0.0 (0) 19.4 (6)

a The POEM question stem is, “Over the past week, on how many nights has your child’s <see first column> because of their eczema?”

Table 4. Concordance between ISAAC and POEM severity measures

 POEM

ISAAC Clear Mild Moderate Severe

Not severe 57.1 (133) 18.5 (43) 21.0 (49) 3.4 (8)

Severe 29.0 (9) 29.0 (9) 35.5 (11) 6.5 (2)

2. Interestingly, the prevalence of disease severity in this cohort was just as high as the ISAAC Phase 3 trials, the Chinese and Japanese studies. However, the AD prevalence was lower than that in HICs. In addition to the statement "AD is known to be multifactorial with genetics and environmental exposures playing important etiologic roles." I would more specifically include how differences in industrialization and the potential to health care access can affect these patterns of prevalence, which may explain the lower prevalences in India and Bangladesh.

We have expanded on this point in lines 279-290 and introduced the idea of the “hygiene hypothesis” which provides evidence that industrialization increases the risk of certain disorders related to immune regulation: 

“Variations in prevalence, both globally and within countries, have yet to be explained. AD is known to be multifactorial with genetics and environmental exposures playing important etiologic roles. The hygiene hypothesis, a widely accepted explanation for the higher prevalence of immunoregulatory disorders in HICs compared to LMICs, suggests that reduced exposure to microbes early in life, such as through better housing, and access to health care, detergents and running water, alters the body’s natural microbiome that helps protect against these disorders, including atopic dermatitis. As Bangladesh continues to see improvements in these sectors, prevalence of such conditions may continue to increase though it is not as straightforward as it once seemed; data regarding seemingly similar exposures is heterogeneous. Childhood helminth exposures, for example, have shown both protective and harmful effects depending on the species [40, 41].”

3. The statement made just before table 2 "Only 10 children - 9 males, 1 female- have ever received a physician's diagnosis of AD" is important to emphasize to the readers and how this differs between other countries that report prevalence of AD with higher rates of physician diagnosed AD.

Although the ISAAC questionnaire has a standard question for doctor’s diagnosis, it is not routinely reported in trial results. Without data based on this standardized question, it is difficult to compare our results with those from other countries. A Nigerian study examined 1,704 children using the ISAAC protocol found an AD prevalence of 10.1% and reported a physician’s diagnosis in 8.7% of those who screened positive. Data from HICs is more difficult to find. A study in Spain among children 10 to 11 years old used the question “Has your son/daughter ever had eczema?” which they considered equivalent to a doctor’s diagnosis ever in the lifetime. For this, 23.4% of parents give affirmative responses and the annual prevalence of atopic dermatitis was 11.4% (Batlles GJ et al. Prevalence and factors linked to atopic eczema in 10- and 11-year-old schoolchildren. Isaac 2 in Almeria, Spain. Allergol Immunopathol (Madr). 2010;38(4):174-180. doi:10.1016/j.aller.2009.10.008). However, because the data in this latter study is not directly comparable to our study or the Nigerian study, we did not include it in the manuscript. We have updated lines 292-297, as follows:

“Notably, only 3.7% of children with AD had ever received formal diagnosis by a physician. This reflects the poor access to healthcare in this population. While the ISAAC questionnaire has a standard question for doctor’s diagnosis, it is not routinely reported in trial results. Data from HICs was not available; however, a study of 1,704 children in Nigeria reported an AD prevalence of 10.1% and that 8.7% had received a formal diagnosis [42].”

Minor points:

1. I would include Line numbers for future manuscript publications to make it easier to reference areas of your manuscript

We have added line numbers to the “clean” draft of our manuscript to make it easier to reference.

2. Table 3 - in the DFI section, align the p-values to all be in the same alignment

Thank you for catching this formatting error. We have updated the table so that all of the p-values are aligned to the right.

Reviewer 3

The study reports interesting data on AD in Bangladesh. It has a large sample size, which in a rural setting may be a real challenge. The authors present a thorough investigation of impact of AD on quality of life in Bangladesh.

Minor comments to the authors

1. In the introduction, authors state that AD prevalence is increasing in low and middle income countries. There is the possibility of actual increase of AD and /or an increase in access to the diagnosis of AD. It would be hard to know whether prevalence is really increasing or health care is now more available and thus cases are being identified. The authors should consider making a statement about this.

We thank Reviewer 3 for highlighting this important point. As access to healthcare increases in LMICs, it is likely that the diagnoses of certain conditions, including atopic dermatitis, will increase. In line 45, we cite Williams et al. (Williams H, Stewart A, von Mutius E, Cookson W, Anderson HR, International Study of A, et al. Is eczema really on the increase worldwide? J Allergy Clin Immunol. 2008;121(4):947-54 e15.) who used data from the ISAAC Phase One and Phase Three trials to conclude that the prevalence of atopic dermatitis is increasing in LMICs. These data were obtained through standardized, validated questionnaires that are independent from ever having had an official clinical diagnosis. They find a 9 fold increase in atopic dermatitis in children age 6 to 7 years old and a 2 fold increase in those 13 to 14 years old. 

We have revised our introductory paragraph to be more thorough and specific. Lines 43 – 47 now read:

“Data from an international study using validated questionnaires found that AD prevalence is increasing in low- and middle-income countries (LMICs), especially among children aged 6 to 7 years old [13]. However, as healthcare access improves in LMICs, increased care-seeking and case detection and reporting may also contribute to higher prevalence estimates.”

2. Discussion: Since children where not evaluated by a health care worker, can we say that the study reports prevalence? Could authors give their view on the ability of ISAAC questionnaire to estimate prevalence? Has the score validated the diagnosis of AD through comparison with actual evaluations by health care workers?

We thank Reviewer 3 for raising these questions. We used ISAAC Phase Three core questionnaire to measure the prevalence of atopic dermatitis. This tool has been validated against a phyisican’s diagnosis of atopic disease in children aged 6-7 and 13-14 years old in multiple countries. It has been used in over 100 countries to measure the prevalence of atopic dermatitis and is the international standard for measuring the prevalence of this disease at a population level. In lines 74 – 80 of the Methods section, we are clear that the ISAAC protocol has not been validated in children under 6 years old which is our study population though it has been used in children as young as 2 years old. 

“We used the International Study of Asthma and Allergies in Childhood (ISAAC) Phase 3 core symptom questionnaire to assess the prevalence and severity of AD since it is the most widely used epidemiologic measure of AD and contains a severity measure. ISAAC has been validated in 6-7 and 13-14 year-old children and has been used but not validated in children as young as 2 years [17]. Our methods of ISAAC administration and comparability to the U.K. Criteria [18-20], an assessment tool that has been validated in infants [21, 22] and young children [20], were reported previously [15].”

However, we also cite our previous work which compared atopic disease prevalence measured by the U.K. Criteria, a tool validated to measure the prevalence of atopic dermatitis in infants and young children, and ISAAC. We found that atopic dermatitis prevalence in children < 1 year was not significantly different between these two tools. Additionally, atopic disease prevalence using the ISAAC protocol was slightly lower in children 1-4 years of age compared to the U.K. Criteria suggesting that our results may underestimate prevalence in this population. While the aim of our study was not to validate ISAAC among a Bangladeshi population, it is an important limitation to highlight and we have expanded upon this in lines 354-364 of our Limitations section as follows:

“Secondly, our research [15] has demonstrated that for children <1 year in this population, AD prevalence as determined by ISAAC matched that of the U.K. Criteria, which has been validated in this age group. However, ISAAC prevalence was lower for children 1-4 years compared to U.K. Criteria, suggesting that our results may be an underestimation of AD prevalence in children 1-4 years. Additionally, while ISAAC is a validated methodology that has been used internationally to measure the prevalence of AD in various races and ethnicities, it has not been validated in Bangladesh. Research from Ethiopia and South Africa that aimed to validate ISAAC and the U.K. Criteria against a physician’s diagnosis suggests that these tools may have low sensitivity in non-Caucasian populations [47, 48].”

Reviewer 4

DEAR EDITOR, I REVIEWED THE ARTICLE ENTITLED AS "Prevalence and psychosocial impact of atopic dermatitis in Bangladeshi children and families" FOR PLOS ONE. IT WAS A WELL WRITTEN ARTICLE DESCRIBING THE EPIDEMIOLOGIC CHARACTERISTICS OF ATOPIC DERMATITIS IN A LOCAL REGION. IT WOULD BE VALUABLE TO PUBLISH THIS LOCAL INFORMATION TO THE WHOLE WORLD.

KIND REGARDS

We thank Reviewer 4 for reviewing our manuscript and for these comments.

We truly thank the Reviewers and Editor for their comments, questions, and suggestions. We believe we have addressed all of their comments with the end result being a greatly improved manuscript. We remain open to further suggestions. 

Again, thank you for your continued consideration and for your thorough review of our manuscript.

Best regards,

Gary Darmstadt

---

## [Editor Report · Decision Letter 1]

26 Mar 2021

Prevalence and psychosocial impact of atopic dermatitis in Bangladeshi children and families

PONE-D-20-34512R1

Dear Dr. Darmstadt,

We’re pleased to inform you that your manuscript has been judged scientifically suitable for publication and will be formally accepted for publication once it meets all outstanding technical requirements.

Kind regards,

Dong Keon Yon, MD, FACAAI

Academic Editor

PLOS ONE

Additional Editor Comments (optional):

I congraturate your amazing paper. Thank you for considering to submit your manuscript in our journal.

Best regards, DK Yon
---

## [Editor Report · Acceptance letter]

8 Apr 2021

PONE-D-20-34512R1 

Prevalence and psychosocial impact of atopic dermatitis in Bangladeshi children and families 

Dear Dr. Darmstadt:

I'm pleased to inform you that your manuscript has been deemed suitable for publication in PLOS ONE. Congratulations! Your manuscript is now with our production department. 

Kind regards, 

on behalf of

Dr. Dong Keon Yon 

Academic Editor

PLOS ONE